# New Insight into a Green Process for Iron Manganese Ore Utilization: Efficient Separation of Manganese and Iron Based on Phase Reconstruction by Vanadium Recycle

**DOI:** 10.3390/ma18040844

**Published:** 2025-02-14

**Authors:** Jing Wen, Xinyu Liu, Shuai Yuan, Tangxia Yu, Lan Zhang, Tao Jiang, Jingwei Li

**Affiliations:** 1School of Metallurgy, Northeastern University, Shenyang 110819, China; wenjing@smm.neu.edu.cn (J.W.); liuxinyu5252@163.com (X.L.); 2210665@stu.neu.edu.cn (T.Y.); 2410749@stu.neu.edu.cn (L.Z.); 2School of Resources and Civil Engineering, Northeastern University, Shenyang 110819, China; 3Key Laboratory for Ecological Metallurgy of Multimetallic Mineral, Ministry of Education, Northeastern University, Shenyang 110819, China; 4Liaoning Key Laboratory for Metallurgical Sensor and Technology, Shenyang 110819, China; 5School of Materials Science and Engineering, Hefei University of Technology, Hefei 230009, China; jwli@hfut.edu.cn

**Keywords:** iron manganese ore, APV roasting, mineral phase reconstruction, manganese vanadate, vanadium recycle

## Abstract

The difficulty of separating iron and manganese is a bottleneck issue in the traditional utilization process of iron manganese ore (Fe-Mn ore). In this work, ammonium polyvanadate (APV), an intermediate product in the vanadium industry, was introduced innovatively to convert the manganese-containing phase in Fe-Mn ore into manganese pyrovanadate (Mn_2_V_2_O_7_) and iron and manganese were then separated efficiently through the acid leaching process. The migration of manganese, iron, and vanadium were systematically studied through XRD, SEM, and leaching experiments. Results show that during the mixed roasting process of Fe-Mn ore and APV, V_2_O_5_, the decomposition product of APV, reacts with the decomposition product of manganese minerals in Fe-Mn ore, Mn_2_O_3_, to produce the target product, acid-soluble Mn_2_V_2_O_7_. Iron and silicon exist in the form of Fe_2_O_3_ and SiO_2_ like in Fe-Mn ore. After the two-step leaching process of the sample roasted at 850 °C with *n*(MnO_2_)/*n*(V_2_O_5_) of 2.25, the leaching ratios of manganese, iron and vanadium are 84.57%, 0.046%, and 4.68%, respectively, achieving the efficient separation of manganese with iron and vanadium. MnCO_3_ obtained by carbonization and precipitation from the manganese-containing leaching solution can be used as an intermediate product of manganese metallurgy or manganese chemical industry. APV obtained by alkaline leaching and precipitation from the vanadium- and iron-containing tailing can be recycled into the roasting system as the roasting additive. The TFe content in the iron-containing tailing reaches 57.21 wt.%, which meets the requirement of iron concentrate. More than 99 wt.% of vanadium from the additive APV can be recovered and recycled back into the Fe-Mn ore utilization process by APV recycling and wastewater recycling, making the Fe-Mn ore utilization with APV roasting a green process.

## 1. Introduction

Manganese, as the typical transition metal, has played critical roles in national economic development and modern national defense construction; it is also an important raw material for development in the fields of new energy and new materials [1,2,3,4]. China is plentiful in the number of deposits in which manganese is present, but it is in a small proportion compared to the other elements [5]. More than 70% of the manganese ore resources in China are ferromanganese ore (Fe-Mn ore), whose mass ratio of manganese to iron is less than 3 [6]. In Fe-Mn ore, iron and manganese with similar physicochemical properties display a symbiotic relationship, which leads to technical difficulties in comprehensive utilization, and Fe-Mn ore has not yet become the main manganese resource for large-scale resource utilization.

The efficient separation of iron and manganese is the core challenge in the resource utilization process of Fe-Mn ore [7,8]. Physical beneficiation, pyrometallurgy, hydrometallurgy, and their combinations are used in the separation process of iron and manganese. Magnetic separation is the process of separating manganese-containing minerals from iron-containing minerals based on the differences in relative magnetic susceptibility among the various metal elements [9,10,11], which is divided into direct magnetic separation and magnetic separation after reduction roasting [12,13]. During the hydrometallurgy processes of Fe-Mn ore, Mn^4+^ is reduce to Mn^2+^ by adding reductants, which are eventually transferred to the manganese-containing solution with some reduced iron [14], resulting in a difficulty in separating iron and manganese from Fe-Mn ore. Many reducing agents have been explored in previous works such as sulfur compounds, organic acid, and inorganic acid [15,16,17,18,19,20,21,22], but they failed to completely separate iron and manganese. Therefore, a highly selective agent is of great significance for Fe-Mn ore to separate iron and manganese.

At this point, some innovative works provided some inspiration. In the vanadium metallurgy industry, Jiang et al. introduced manganese carbonate or manganese dioxide as additives into the roasting process of vanadium slag for the extraction of vanadium [23,24]. Manganese selectively converted vanadium spinel into the acid-soluble manganese vanadate, and manganese was reduced during this process. However, iron in vanadium slag was difficult to reduce, thus achieving the separation of manganese and iron during the leaching process. Looking at it from a different perspective, it is worth studying whether vanadium can serve as a medium for separating manganese and iron in the extraction process of Fe-Mn ore.

Hence, in this work, in order to improve the separation efficiency of iron and manganese in Fe-Mn ore, a novel additive, ammonium polyvanadate (APV), is introduced to the roasting process of Fe-Mn ore. APV is the intermediate product in the vanadium industry, which is rich in vanadium and can selectively convert the manganese-containing phase in Fe-Mn ore into the acid soluble manganese pyrovanadate (Mn_2_V_2_O_7_). But the iron-containing phase does not react with vanadium, resulting in the efficient separation of iron and manganese. In addition, in the subsequent leaching and vanadium extraction processes, APV can be further recovered and recycled, making the utilization of Fe-Mn ore more efficient and greener. In this work, manganese, iron, and vanadium, as the three important components, their phase reconstruction rule during roasting, and their separation behaviors during leaching are systematically studied. Finally, they are recovered and recycled in the different products.

## 2. Material and Methods

### 2.1. Material

Fe-Mn ore and APV were acquired from Weixin Manganese Products Co., Ltd., Hunan, China and Xichang Steel Vanadium Co., Ltd., Sichuan, China respectively. H_2_SO_4_ (Sinopharm Chemical Reagent Co., Ltd., Beijing, China, analytical purity > 98 wt.%) was the leaching medium during two-step H_2_SO_4_ leaching and was used to adjust the pH value during vanadium precipitation. NaOH (Sinopharm Chemical Reagent Co., Ltd., Beijing, China, analytical purity > 98 wt.%) was the medium during alkaline leaching for vanadium extraction from vanadium- and iron-containing tailings and was used to adjust the pH value during manganese precipitation. (NH_4_)_2_SO_4_ (Sinopharm Chemical Reagent Co., Ltd., Beijing, China, analytical purity > 99 wt.%) was used to precipitate vanadium in the form of APV.

### 2.2. Procedure

#### 2.2.1. Mixed Roasting

APV was used as the roasting addictive in this work. The molar ratio of MnO_2_ in Fe-Mn ore and V_2_O_5_ in APV, represented as *n*(MnO_2_)/*n*(V_2_O_5_), was used to represent the mixing ratio of Fe-Mn ore and APV, which ranged from 0 to 3. As a comparison, the experiment of roasting Fe-Mn ore individually without adding APV was also carried out. Putting the mixed material into the muffle furnace at room temperature, it was then heated to the specific temperature at a rate of 5 °C/min, and held for 180 min. The roasting temperature ranges from 300 °C to 950 °C. The experimental parameters are shown in Table 1. The roasted samples were finally ground to less than 74 μm to investigate the phases, element distribution, and leaching behaviors.

#### 2.2.2. Two-Step H_2_SO_4_ Leaching

The leaching process is caried out to achieve the dissolution of manganese vanadate, the separation of iron and manganese, as well as the separation of manganese and vanadium. The leaching process consisted of two consecutive stages and was carried out in the same device. In the first stage, a sulfuric acid system at a pH value of around 2.2 was used as the leaching media to dissolve manganese vanadate effectively and achieve the separation of iron and manganese. The pH value of 2.2 is considered to be the most favorable for the dissolution of manganese vanadate [25], and the reaction equation is shown in Equation (1). During leaching, the roasted materials were put in the deionized water with a ratio of liquid to solid of 5 (L/S = 5; mL/g) until the deionized water was heated to 60 °C. Then, the pH value of the leaching system was kept at around 2.2 for 60 min by adding 10 vol.% H_2_SO_4_ continuously.Mn_2_V_2_O_7_ + 6 H^+^ = 2(VO_2_)^+^ + 2 Mn^2+^ + 3 H_2_O(1)

In the second stage, the pH value of leaching was adjusted to 1.8 to promote the precipitation of vanadium in the leaching solution, as shown in Equation (2) [25], thereby achieving the separation of vanadium and manganese further. A measure of 10 vol.% of H_2_SO_4_ was added to the leaching system to keep the pH value at around 1.8 for 60 min. After the two-step leaching, the manganese-containing leaching solution and the vanadium- and iron-containing tailing, were collected by filtration. In order to investigate the extraction behavior of manganese, as well as the migration behavior of iron and vanadium as impurity components, the leaching ratios of manganese, iron, and vanadium were investigated, which were calculated with Equation (3).2(VO_2_)^+^ + H_2_O = V_2_O_5_ ↓ + 2 H^+^
(2)Leaching ratio (%) = (*m*_L_/*m*_0_) × 100(3)
where *m* represents manganese, iron, or vanadium; *m*_L_ is the total mass of *m* in the leaching solution, *m*_0_ is the total mass of *m* in the roasted materials.

#### 2.2.3. Manganese Recovery

Carbonization was applied to recover manganese in the leaching solution. CO_2_ was injected into the manganese-containing vanadium wastewater for 30 min at a flow rate of 0.5 L/min. NaOH solution was added to keep the pH value of the precipitation reaction at 7.0. The above experimental conditions were conducted in reference to the previous studies [24]. Manganese products and the wastewater left after manganese precipitation were collected through filtration.

#### 2.2.4. APV Recycle

Vanadium in the additive APV was transferred into the tailing, which was leached under the alkaline condition to extract vanadium and achieve the separation of vanadium and iron further. The tailing was put in the deionized water with a ratio of liquid to solid of 5 (L/S = 5; mL/g) at room temperature. Then, the pH value of the leaching system was kept at around 12.0 for 60 min by adding NaOH solution continuously. After vanadium leaching, the leaching solution containing vanadium and the tailing containing iron were collected by filtration. Vanadium in the leaching solution was further extracted by precipitation with (NH_4_)_2_SO_4_ and recycled to the next roasting process in the form of APV. The mass ratio of (NH_4_)_2_SO_4_ to vanadium in the leaching solution (*m*_(NH4)2SO4_/*m*_v_) was 1.5. In addition, the precipitation time, precipitation temperature, and pH value were set as 60 min, 95 °C, and around 2.0, respectively, following our previous work [26,27]. After the vanadium precipitation process, the wastewater could be recycled back into the acid leaching system. In addition, the wastewater after manganese precipitation in Section 2.2.3 could be used as the medium for alkaline leaching of vanadium to ensure that the trace amount of vanadium in the wastewater is not lost. The whole process is presented in Figure 1, where the three circulation routes mentioned above are also recorded.

### 2.3. Characterization

The chemical compositions of Fe-Mn ore, APV, produced MnCO_3_, vanadium- and iron-containing tailing, and iron-containing tailing were detected using ICP–AES (PerkinElmer Optima–4300DV, Perkin Elmer Enterprise Management (Shanghai) Co., Ltd., Shanghai, China). X-ray Diffraction Analysis (XRD, X’PERT PRO MPD/PW3040, PANalytical B.V. Corporation, Almelo, The Netherlands, Cu Kα radiation, *λ* = 0.15406 nm, 2θ = 10°–80° in steps of 0.013°, database for phase identification is International Crystal Structure Database, ICSD) and a scanning electron microscope (SEM, Ultra Plus, Zeiss, Jena, Germany) equipped with Energy Disperse X-ray Spectrometry (EDS, INCA Energy 350, Zeiss, Germany) were used to characterize the phases of solid samples containing Fe-Mn ore, APV, the roasted materials, and tailings, along with the products containing manganese and vanadium. The mass of vanadium in the leaching solution was analyzed using the ferrous ammonium sulfate titration method [28]. The concentration of manganese and iron in the leaching solution was detected by the Atomic Absorption Spectrophotometer (TAS-990, Beijing, China).

## 3. Results

### 3.1. Composition and Microstructure of Fe-Mn Ore and APV

The chemical composition of Fe-Mn ore in this work is shown in Table 2. The contents of MnO_2_ and Fe_2_O_3_ are 26.97 wt.% and 66.27 wt.%, and the mass ratio of TMn and TFe is 0.37, far lower than 3, meaning that the raw material is a typical Fe-Mn ore. Its XRD pattern and SEM image with element mapping are shown in Figure 2. Results show that iron and manganese are almost present in the different phases as the blocky particles in Figure 2b. Their respective existence areas almost do not overlap according to Figure 2c,d, with hematite (Fe_2_O_3_ -JCPDS card no. 01-084-0306) being the most predominant iron-containing phase. Manganese exists in pyrolusite (MnO_2_ -JCPDS card no. 01-072-1982) and manganese silicate (Mn_7_SiO_12_ -JCPDS card no. 00-033-0904) phases, which is confirmed by the overlapping distribution areas of some manganese and silicon in Figure 2c,e. Other silicon appears in the form of quartz (SiO_2_ -JCPDS card no. 00-046-1045). These phases are consistent with their standard cards. In addition, APV, with the chemical formula of (NH_4_)_2_V_6_O_16_, from Panzhihua Iron and Steel Company was used as the additive to reconstruct the manganese-containing phase in Fe-Mn ore into the acid-soluble manganese vanadate, whose XRD pattern and SEM image with EDS analysis are also shown in Figure 2. The diffraction peaks of APV are in complete coincidence with those of the standard card according to JCPDS 01-087-0603. The micrograph shown in Figure 2f demonstrates that APV has a layered structure and is concentrated on a large amount of vanadium according to Figure 2g.

### 3.2. Phase Reconstruction During Roasting Process

#### 3.2.1. Effect of *n*(MnO_2_)/*n*(V_2_O_5_) on Phase Reconstruction

The efficient separation of iron and manganese in Fe-Mn ore through the selective combination of manganese and vanadium is the substantial difference that differentiates this work from other methods; therefore, the addition amount of APV is studied as the primary influencing factor. Figure 3 shows the XRD patterns, SEM images, and element distributions of the products roasted at 850 °C with different *n*(MnO_2_)/*n*(V_2_O_5_). When Fe-Mn ore is roasted without APV at 850 °C, Fe_2_O_3_ and SiO_2_ still exist in the roasted Fe-Mn ore, while the diffraction peaks of MnO_2_ and Mn_7_SiO_12_ are difficult to detect, replaced by the diffraction peaks of FeMnO_3_. This means that roasting Fe-Mn ore without addictive APV could only convert Mn(IV) into Mn(III), and iron would react with manganese to form FeMnO_3_, which is difficult to be dissolved in the acid solutions with low concentration. When APV is added to Fe-Mn ore for roasting, iron and silicon mainly exist in the form of Fe_2_O_3_ and SiO_2_, respectively, while the phase of manganese changes with the amount of APV added. When *n*(MnO_2_)/*n*(V_2_O_5_) is 0.5, 1, and 1.5, manganese and some iron react with vanadium and form Fe_4_Mn_3_(VO_4_)_6_. The element distribution and EDS analysis of the roasted sample at *n*(MnO_2_)/*n*(V_2_O_5_) of 1 could also prove this conclusion. Many of the distribution areas of vanadium, manganese, and iron overlap, and the EDS analyses at points 1 to 4 in these overlapping areas show that the atomic ratios of iron, manganese, and vanadium are around 4:3:6, which is consistent with that in Fe_4_Mn_3_(VO_4_)_6_. In addition, iron that does not overlap with vanadium and manganese is considered to be Fe_2_O_3_, as demonstrated by the EDS analysis at point 5.

When *n*(MnO_2_)/*n*(V_2_O_5_) is 2 and 2.25, manganese and iron in the roasted samples no longer exist in the same phases, and vanadium reacts with manganese to form Mn_2_V_2_O_7_, whose diffraction peaks are consistent with its standard card according to JCPDS 00-052-1266. The chemical equations for the reactions between manganese-containing phases in Fe-Mn ore and APV are shown in Equations (4) and (5). Iron and manganese in the roasted samples are Fe(III) and Mn(II), respectively, meaning that the addition of APV further reduce Mn(III) to Mn(II) selectively. According to the element mapping of the roasted sample at *n*(MnO_2_)/*n*(V_2_O_5_) of 2.25, the distribution areas of manganese and vanadium are almost the same, while they are completely different from those of iron and silicon. In the EDS analyses of points 6 to 8, the atomic ratios of manganese and vanadium are around 1, which is consistent with that in Mn_2_V_2_O_7_. In addition, point 9 and point 10 correspond to Fe_2_O_3_ and SiO_2_, respectively.

When *n*(MnO_2_)/*n*(V_2_O_5_) is 2.5 and 3, Fe_2_O_3_, SiO_2_, and Mn_2_V_2_O_7_ are still the main phases in the roasted sample. In addition, some manganese reacts with iron to form FeMnO_3_. This is because APV added at this *n*(MnO_2_)/*n*(V_2_O_5_) value is not sufficient to react all manganese into Mn_2_V_2_O_7_, so the remaining manganese will react with iron to form FeMnO_3_. At a *n*(MnO_2_)/*n*(V_2_O_5_) of 3, the element mapping of iron and manganese in the roasted samples partially overlap, and point 11 and point 12, as the particles of Mn_2_V_2_O_7_, have significantly higher values of iron content compared to a *n*(MnO_2_)/*n*(V_2_O_5_) of 2. Point 13 and point 14 correspond to Fe_2_O_3_, while point 15 has a lower content of all elements except for manganese, indicating the presence of an obvious manganese-containing phase that has not been reacted with vanadium under *n*(MnO_2_)/*n*(V_2_O_5_) of 3.6 MnO_2_ + (NH_4_)_2_V_6_O_16_ = 3 Mn_2_V_2_O_7_ + 2 NH_3_↑+ H_2_O↑ + 3 O_2_↑(4)6 Mn_7_SiO_12_ + 7 (NH_4_)_2_V_6_O_16_ = 21 Mn_2_V_2_O_7_ + 6 SiO_2_ + 14 NH_3_↑+ 7 H_2_O↑ + 9 O_2_↑ (5)

#### 3.2.2. Effect of Roasting Temperature on Phase Reconstruction

Roasting temperature is a significant factor in the roasting process for the phase evolution, hence the effect of roasting temperature on the products roasted at *n*(MnO_2_)/*n*(V_2_O_5_) of 2.5 is investigated by XRD and SEM, which are shown in Figure 4. In the products obtained at the different roasting temperatures, both Fe_2_O_3_ and SiO_2_, present stably, indicating that the introduction of APV has almost no effect on the iron and silicon components in Fe-Mn ore. At 300 °C to 400 °C, the diffraction peak of APV around 28° is difficult to observe, and the diffraction peaks of its decomposition product V_2_O_5_ appear, indicating that APV gradually decomposes and produces V_2_O_5_ with increasing roasting temperature according to Equation (6):(NH_4_)_2_V_6_O_16_ = 3V_2_O_5_ + 2NH_3_↑ + H_2_O(6)

MnO_2_ and Mn_7_SiO_12_ are still the main manganese-containing phases. At 500 °C, the diffraction peaks of MnO_2_ are replaced by those of Mn_2_O_3_, indicating that MnO_2_ gradually decomposes to form Mn_2_O_3_, as shown in Equation (7):4MnO_2_ = 2Mn_2_O_3_ + O_2_↑(7)

In addition, diffraction peaks of Fe_4_Mn_3_(VO_4_)_6_ are also observed, indicating that Mn_2_O_3_, V_2_O_5_, and Fe_2_O_3_ have begun to react at 500 °C, according to Equation (8):6Mn_2_O_3_ + 12V_2_O_5_ + 8Fe_2_O_3_ = 4Fe_4_Mn_3_(VO_4_)_6_ + 3O_2_↑(8)

This has also been confirmed by the finding that manganese, vanadium, and iron partially overlap in the same area according to the element mapping of the roasted sample at 500 °C. In the EDS analysis results of point 1, the simultaneously high contents of manganese, vanadium, and iron also proves this conclusion. In addition, most of manganese, vanadium, silicon, and iron are still present in the different regions as Mn_2_O_3_, V_2_O_5_, SiO_2_, and Fe_2_O_3_ respectively, which is consistent with the ESD analyses of point 2 to point 5.

When the roasting temperature is increased to 600 °C, the diffraction peaks of Mn_7_SiO_12_, Mn_2_O_3_, and V_2_O_5_ cannot be detected. More Fe_4_Mn_3_(VO_4_)_6_ is generated than at 500 °C. At 700 °C, the intensity of the diffraction peaks of Fe_4_Mn_3_(VO_4_)_6_ in the roasted sample is significantly reduced; correspondingly, the diffraction peaks of Mn_2_V_2_O_7_ are clearly observed according to Equation (9):2Mn_2_O_3_ + 2V_2_O_5_ = 2Mn_2_V_2_O_7_ + O_2_↑(9)

The element mapping of the sample at 700 °C further indicates that small amounts of iron, vanadium, and manganese are concentrated in the same area where point 6 is located. Most of the manganese only has overlapping areas of distribution with vanadium, and the atomic ratio of vanadium to manganese in the EDS analysis of points 7 and 8 is around 1, confirming the generation of Mn_2_V_2_O_7_. The EDS analyses of point 9 and point 10 indicate that these two points correspond to Fe_2_O_3_. When the roasting temperature is increased from 800 °C to 950 °C, the manganese-containing phase in the roasted sample stabilizes with Mn_2_V_2_O_7_. The other phases are not significantly different. The element distribution of the roasted sample at 850 °C shows that the vanadium and manganese are almost completely overlapped with each other and are separate from iron. The above study shows that V_2_O_5_ generated from the decomposition of APV during roasting can selectively react with Mn_2_O_3_ generated from the decomposition of the manganese-containing phase in Fe-Mn ore to generate Mn_2_V_2_O_7_, while the iron-containing phase is still dominated by Fe_2_O_3_, which realizes the high-efficiency separation of iron and manganese.

### 3.3. Separation Behaviors of Components During Two-Step Leaching

#### 3.3.1. Effect of *n*(MnO_2_)/*n*(V_2_O_5_) on Leaching Behaviors of Components

After the two-step leaching, the leaching ratio of manganese and the mass fraction of manganese in the tailing of the roasted samples under different *n*(MnO_2_)/*n*(V_2_O_5_) are shown in Figure 5a. After leaching the sample obtained from individually roasted Fe-Mn ore without APV, the leaching ratio of manganese is only 16.16%, and the mass fraction of manganese in the tailings is still 15.50%, which indicates that it is difficult to realize the manganese extraction from the roasted Fe-Mn ore alone. When APV is used as the roasting additive, with the increase in *n*(MnO_2_)/*n*(V_2_O_5_) from 0 to 3, the leaching ratio of manganese exhibits first elevation and then reduction, which reaches the maximum value of 84.57% when *n*(MnO_2_)/*n*(V_2_O_5_) is 2.25; the mass fraction of manganese in the tailings is only 2.67% at this time. This variation is consistent with the generation of the target product Mn_2_V_2_O_7_ in the roasted sample. When *n*(MnO_2_)/*n*(V_2_O_5_) is less than 1.5, due to the formation of acid insoluble phase of Fe_4_Mn_3_(VO_4_)_6_, the leaching ratio of manganese is low. With the gradual increase in *n*(MnO_2_)/*n*(V_2_O_5_) to 2.25, Mn_2_V_2_O_7_ with good acid solubility is gradually generated, resulting in a high leaching ratio of manganese. When continuing to increase *n*(MnO_2_)/*n*(V_2_O_5_), the amount of APV added is not sufficient to completely convert the manganese-containing phase into Mn_2_V_2_O_7_. Other manganese-containing phases that do not exist in the form of Mn_2_V_2_O_7_ are difficult to dissolve in the sulfuric acid solutions or to achieve the leaching of manganese.

In addition to manganese, the leaching ratios of iron and vanadium during the leaching process are shown in Figure 5b. The leaching ratio of iron is always lower than 0.4%, and even lower than 0.2% when APV is added. When the leaching ratio of manganese reaches the maximum value of 84.57%, the leaching ratio of iron is only 0.046%, achieving the efficient separation of iron and manganese. Furthermore, the leaching ratio of vanadium shows an overall increasing trend, but it is always lower than 6%. When *n*(MnO_2_)/*n*(V_2_O_5_) is 2.25, the leaching ratio of vanadium is only 4.68%, achieving the efficient separation of manganese and vanadium. This is because in the second stage of leaching, most of vanadium that has already been leached in the first stage undergoes the hydrolysis and precipitation reaction.

#### 3.3.2. Effect of Roasting Temperature on Leaching Behaviors of Components

The leaching ratio of manganese and the mass fraction of manganese in the tailing of the roasted samples under different roasting temperatures are shown in Figure 6a. The leaching ratio of manganese first increases and then decreases when the roasting temperature is gradually increased, and, correspondingly, the mass fraction of manganese in the tailings after acid leaching decreases and then increases. When the roasting temperature is lower than 700 °C, the leaching ratio of manganese is lower than 20% because almost no acid soluble Mn_2_V_2_O_7_ is generated at low temperatures. After the roasting temperature is higher than 700 °C, the manganese leaching ratio increase significantly to 59.12%, 77.00%, and 84.57% at 700 °C, 800 °C, and 850 °C, respectively, which corresponds to the increase in target product Mn_2_V_2_O_7_ with the increase in roasting temperature. Continuing to increase the roasting temperature to 900 °C and 950 °C, the high roasting temperature leads to the further transformation of the manganese-containing phase, resulting in the slight decrease of manganese leaching ratio.

In addition to manganese, the leaching ratios of iron and vanadium during the leaching process are shown in Figure 6b. The trends of leaching ratios of iron and vanadium are almost consistent with that of manganese. When the roasting temperature is 850 °C, at which the leaching ratio of manganese reaches the maximum value of 84.57%, the leaching ratios of iron and vanadium are 0.046% and 4.68%, achieving the separation of manganese with iron and vanadium.

### 3.4. Manganese Recovery

The manganese ions in the acid leaching solution are recovered in the form of manganese by carbonization and precipitation. CO_2_ as the carbonizing agent is converted to CO_3_^2−^ in the alkaline solution as shown in Equation (8) and then combined with manganese ions to form MnCO_3_, according to Equation (9). After carbonization and precipitation, more than 99 wt.% of manganese ions are extracted from the leaching solution in the form of MnCO_3_.CO_2_ + 2OH^−^ = CO_3_^2−^ + H_2_O(10)Mn^2+^ + CO_3_^2−^ = MnCO_3_↓(11)

Figure 7 shows the XRD pattern and microscopic morphology of the generated MnCO_3_. In Figure 7a, the diffraction peaks of the generated MnCO_3_ are consistent with its standard card (JCPDS: 00-007-0268), and there are no obvious impurity diffraction peaks, which proves that the MnCO_3_ prepared is of high purity. In the SEM images of different magnifications in Figure 7b,c, MnCO_3_ is shown as the spherical particle with a smooth surface and a diameter of less than 1 μm. According to the element mapping and EDS analysis of point A in Figure 7d,e, in addition to a large amount of manganese, there is a small amount of vanadium and sodium in the generated MnCO_3_, which are, respectively, from the manganese-containing leaching solution and NaOH solution used to adjust the pH value during carbonization.

Furthermore, according to the ICP-AES analysis, the purity of the generated MnCO_3_ is greater than 95 wt.%, and the mass fractions of impurities vanadium and sodium are 1.03% and 1.15%, respectively. The quality of the generated MnCO_3_ is even higher than the requirements of the first-grade industrial manganese carbonate (HG/T 4203-2011). This MnCO_3_ product can be used as an industrial raw material in the fields of manganese metallurgy or manganese chemistry.

### 3.5. APV Recycle

After leaching, more than 99.9 wt.% of iron and 95 wt.% of vanadium in the roasted sample remain in the tailing, which is called the vanadium- and iron-containing tailing, whose chemical composition is shown in Table 3. Its micro-morphology and major element distributions are shown in Figure 8. The tailing particles are present as flocculent or blocky structures according to the SEM images with different magnifications in Figure 8a,b. The flocculent particles distribute randomly on the blocky particles. According to the element mapping of the main elements in Figure 8c,f, almost no manganese could be detected, iron and vanadium are the main components in the tailings, meaning that the efficient separation of manganese with iron and vanadium has been realized. Furthermore, the blocky particles are mainly Fe_2_O_3_ and SiO_2_, and a large amount of vanadium accumulates on the flocculent particles, which is V_2_O_5_ produced through hydrolysis during the second leaching process.

Almost all of the vanadium in the vanadium and iron tailing is transferred to the leaching solution during alkaline leaching, and then vanadium can be recovered by ammonium precipitation and form APV. Figure 9 demonstrates the XRD pattern and microscopic morphology of the generated APV. In Figure 9a, the intensity and position of the diffraction peaks of the precipitated product are consistent with the standard card of APV (JCPDS: 01-087-0603). From Figure 9b,c, it can be seen that the precipitated APV shows a layered structure, which is consistent with the microscopic morphology of the roasting additive APV in Figure 2. The EDS analysis of point B in Figure 9e does not show the existence of other impurity elements, proving the high purity of the obtained APV. Moreover, the chemical analysis results show that the purity of the precipitated APV is close to 100%, even higher than that of the APV used as the original roasting additive in 2.1, which can be used as a subsequent roasting additive for Fe-Mn ore, realizing the recycling of the additive APV.

### 3.6. Iron Utilization

After the alkaline leaching of vanadium- and iron-containing tailing, almost all the iron remains in the tailings, which are called iron-containing tailings, whose chemical composition is shown in Table 4. Results show that the TFe content of the iron-bearing tailings is 57.21 wt.%, which is nearly 11 percentage points higher than the TFe content of Fe-Mn ore (Table 2). The TFe content has reached the national standard for iron concentrate, and the content of Na_2_O and SiO_2_ also meets the national standard requirements, so the iron-containing tailing can be utilized directly as a smelting raw material. Its physical phase composition, microscopic morphology, and distribution of main elements are shown in Figure 10. The results show that the main phases of iron-containing tailing are Fe_2_O_3_ and SiO_2_. Compared with Figure 8, the flocculent particles disappear under the microscopic field of view, which is due to the dissolution of vanadium hydrolysis products during the alkaline leaching process.

## 4. Evolution

### 4.1. Efficient Separation of Iron and Manganese

The most notable difficulty in the utilization of Fe-Mn ore is the efficient separation of iron and manganese, which has been a key concern of researchers in recent decades. In our work, the strong reactivity of manganese with vanadium is utilized to selectively extract manganese by adding APV, resulting in the significant separation efficiency of iron and manganese. To evaluate and compare the separation behaviors of iron and manganese better among the different research reports, the ratios of manganese leaching and iron leaching are calculated and displayed in Figure 11. The higher the ratio, the better the separation effect of iron with manganese. As can be seen from Figure 11, the ratio of manganese leaching and iron leaching has been reported to be less than 250, while that value in this work is even more than 1800, which indicates that the separation of iron and manganese is very desirable [10,16,18,20,21,22,29,30,31,32,33,34].

### 4.2. Three Circular Routes for Vanadium Recycle

In this work, a novel and efficient process for Fe-Mn ore utilization is proposed by APV roasting. During this process, three circular routes are proposed, which not only save the production costs but also avoid the discharge of industrial waste.

Firstly, the roasting additive, APV, reacts with the manganese-containing phase in the Fe-Mn ore to form Mn_2_V_2_O_7_ during roasting, and then, vanadium is transferred to the vanadium and manganese containing tailing during the leaching process. After alkaline solution leaching and ammonium salt precipitation, vanadium is transformed into APV. The newly generated APV has a higher purity than the industrial grade intermediate, and can be recycled in the roasting process.

Secondly, after the carbonation and MnCO_3_ preparation process, there is still a small amount of vanadium ions in the wastewater after manganese recovery, which can be used as a medium for the alkaline leaching process of vanadium- and iron-containing tailing, and the vanadium remaining in the waste can be extracted in the following precipitation process.

Thirdly, after vanadium recovery by precipitation with (NH_4_)_2_SO_4_, the wastewater can be recycled as the leaching medium during sequential extraction with some deionized water added to complement. As a result of these three recycling routes, the loss of vanadium from additive APV is concentrated in the iron-containing tailing. Hence, more than 99 wt.% of vanadium in the additive APV can be recovered and recycled back into the process of Fe-Mn ore utilization, which can be confirmed by the finding that the content of vanadium in the iron-containing tailing is only 0.50 wt.%. In summary, the process route reported in this work can achieve both the recycling of additives and the effective reduction in industrial wastewater discharge. Figure 12 shows the process flow sheet of Fe-Mn ore utilization based on APV roasting in this work.

## 5. Conclusions

In this work, APV, an intermediate product in the vanadium industry, was innovatively introduced to the roasting process of Fe-Mn ore. A green process for the resource utilization of Fe-Mn ore has been proposed, and the efficient separation of iron and manganese has been achieved. The specific conclusions are as follows.

(1)When Fe-Mn ore and APV are mixed with *n*(MnO_2_)/*n*(V_2_O_5_) of 2.25, and roasted at 850 °C for 120 min, the decomposition product V_2_O_5_ of APV reacts with the decomposition product Mn_2_O_3_ of MnO_2_ and Mn_7_SiO_12_ in Fe-Mn ore to produce the target product, acid soluble Mn_2_V_2_O_7_. Fe_4_Mn_3_(VO_4_)_6_ is an important intermediate product in the roasting process, which is generated under conditions of lower roasting temperature and lower *n*(MnO_2_)/*n*(V_2_O_5_) value. Iron and silicon exist in the form of Fe_2_O_3_ and SiO_2_, respectively.(2)During the two-step acid leaching, the target product Mn_2_V_2_O_7_ in the roasted sample is dissolved to achieve the separation of manganese and iron in the first leaching step, and vanadium is further hydrolyzed and precipitated to achieve the separation of manganese and vanadium in the second leaching step. When the leaching ratio of manganese reaches the maximum value of 84.57%, the leaching ratios of iron and vanadium are only 0.046% and 4.68%, it achieves the efficient separation of manganese, iron, and vanadium. The leaching ratio of manganese is consistent with the generation laws of Mn_2_V_2_O_7_ during the roasting process(3)Manganese ions in the manganese-containing leaching solution are crystallized in the form of MnCO_3_ by carbonization and can be used as an intermediate product in the manganese metallurgy or manganese chemical industry. Vanadium in the vanadium- and iron-containing tailing is received by alkali leaching and precipitation to obtain APV, which can be recycled back to the Fe-Mn ore roasting as an additive. The TFe grade of iron-containing tailing has increased from 46.39% to 57.21%, which has met the requirement for iron content in the iron concentrate.(4)The separation efficiency of iron and manganese reported in this work is much higher than the results reported in the current literature. During this process, three circular routes are proposed, including the suggestions that APV could be recycled in the roasting process, the manganese extraction wastewater could be recycled for the alkaline leaching process, and the manganese extraction wastewater could be recycled for the acid leaching process. More than 99 wt.% of vanadium in the additive APV can be recovered and recycled back into the process of Fe-Mn ore utilization by APV and wastewater recycling.

## Figures and Tables

**Figure 1 materials-18-00844-f001:**
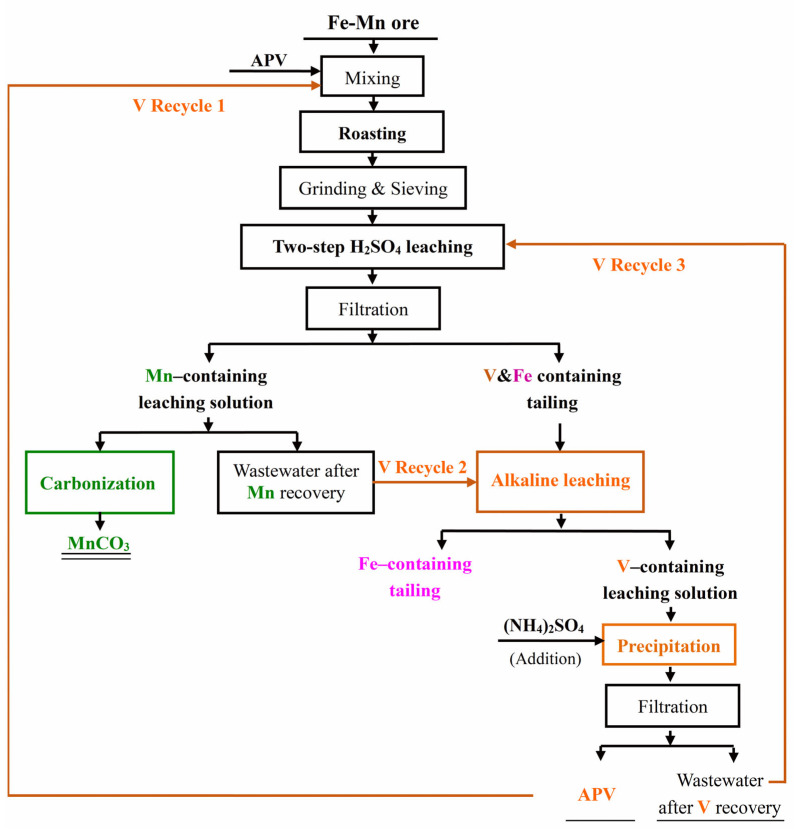
Flow chart of comprehensive utilization of Fe-Mn ore.

**Figure 2 materials-18-00844-f002:**
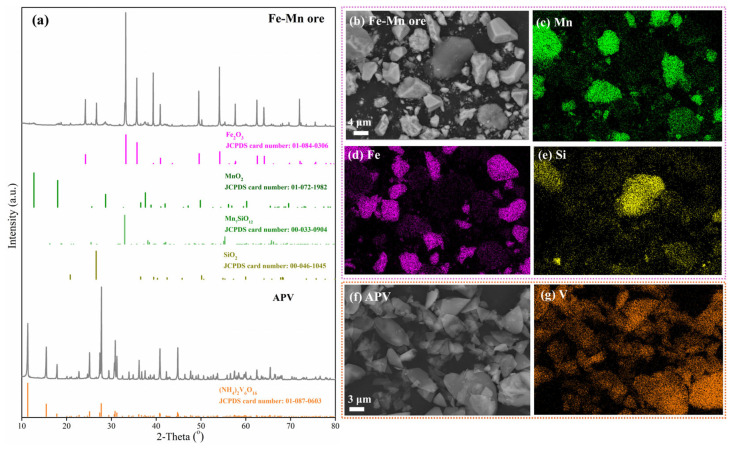
Characterization of Fe-Mn ore and APV: (**a**) XRD pattern; (**b**–**g**) SEM image with elements mapping.

**Figure 3 materials-18-00844-f003:**
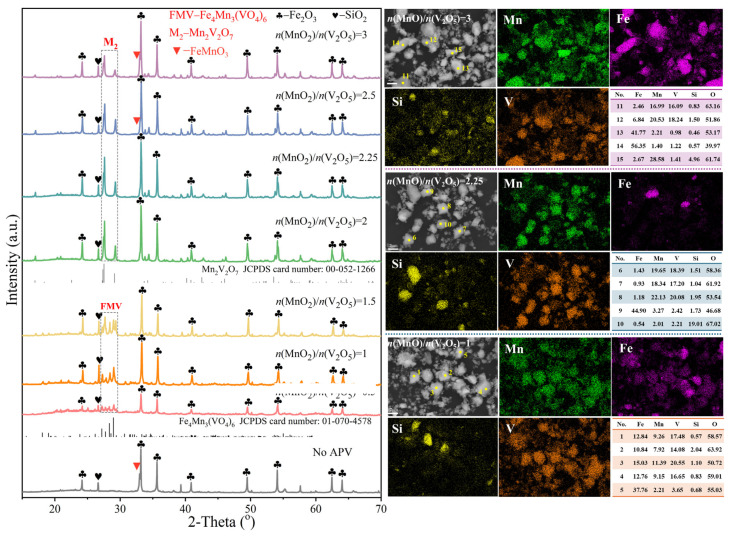
XRD and SEM of the roasted materials under different amount of APV.

**Figure 4 materials-18-00844-f004:**
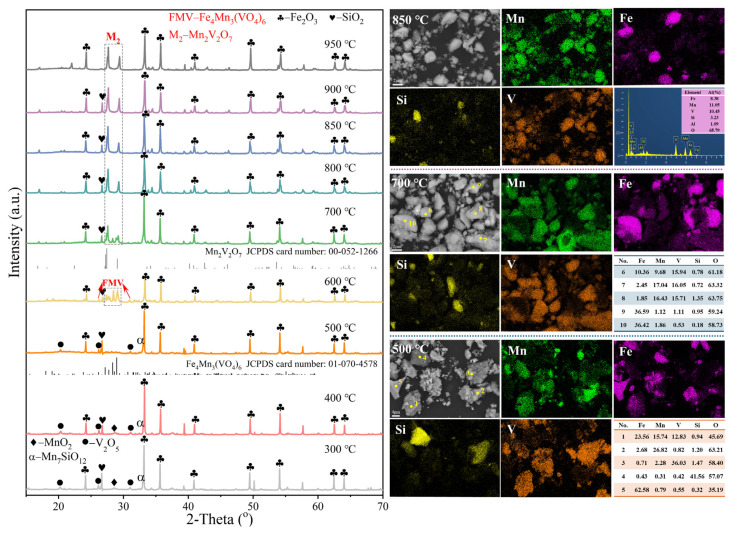
XRD of the roasted samples under different roasting temperature.

**Figure 5 materials-18-00844-f005:**
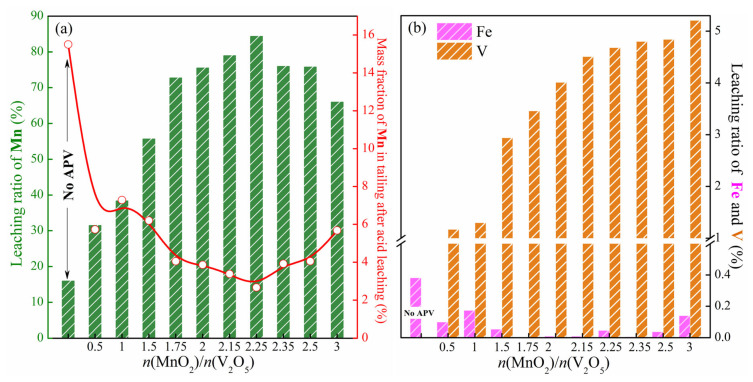
Effect of *n*(MnO_2_)/*n*(V_2_O_5_) on the leaching behaviors: (**a**) leaching ratio of manganese (green bars) and mass fraction of manganese in tailing after acid leaching (red line and circles), (**b**) leaching ratio of iron and vanadium.

**Figure 6 materials-18-00844-f006:**
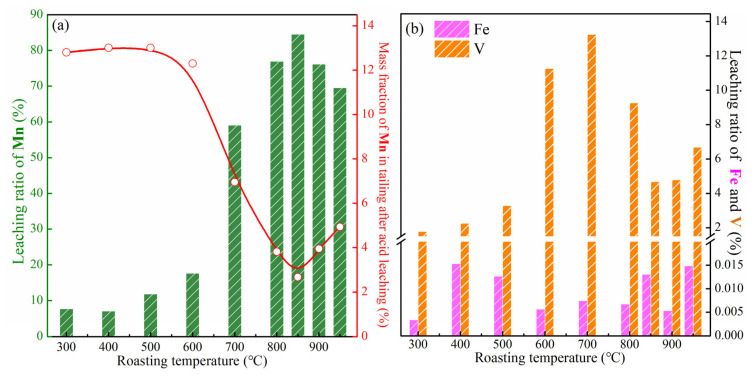
Effect of roasting temperature on the leaching behaviors: (**a**) leaching ratio of manganese (green bars) and mass fraction of manganese in tailing after acid leaching (red line and circles), (**b**) leaching ratio of iron and vanadium.

**Figure 7 materials-18-00844-f007:**
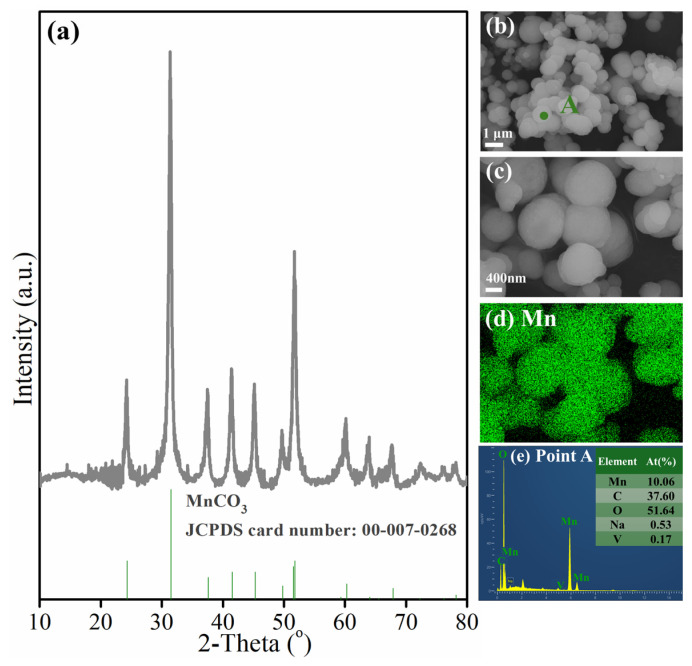
Characterization of precipitation product MnCO_3_: (**a**) XRD pattern; (**b**–**d**) SEM image with elements mapping; (**e**) EDS analysis of point A in (**b**).

**Figure 8 materials-18-00844-f008:**
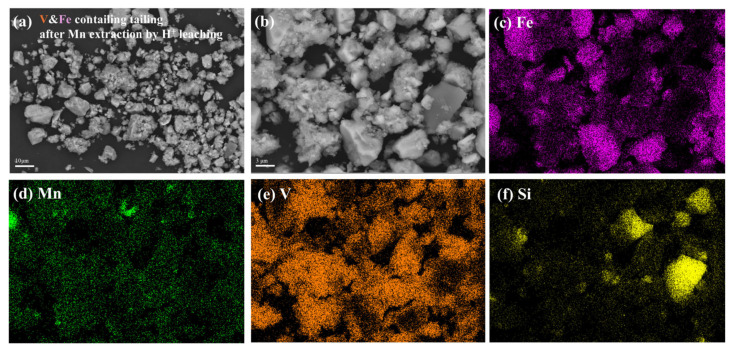
Characterization of vanadium- and iron-containing tailing: (**a**,**b**) morphology; (**c**–**f**) elements distribution.

**Figure 9 materials-18-00844-f009:**
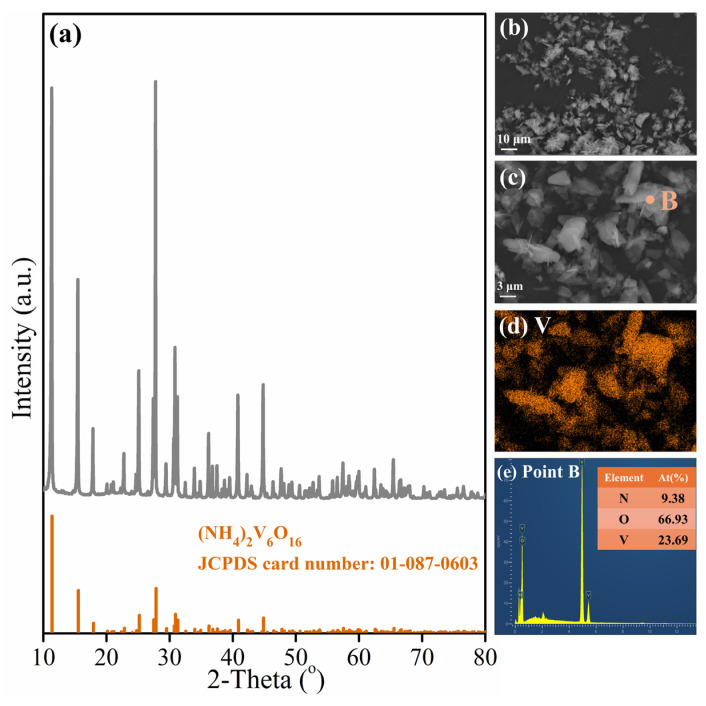
Characterization of precipitated APV: (**a**) XRD pattern; (**b**–**d**) morphology and elements distribution; (**e**) EDS analysis of point B in (**c**).

**Figure 10 materials-18-00844-f010:**
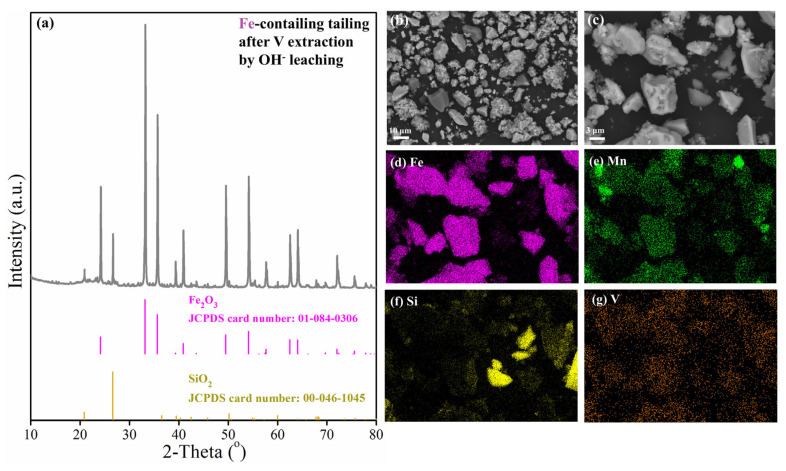
Characterization of iron-containing tailing: (**a**) XRD pattern; (**b**–**g**) morphology and elements distribution.

**Figure 11 materials-18-00844-f011:**
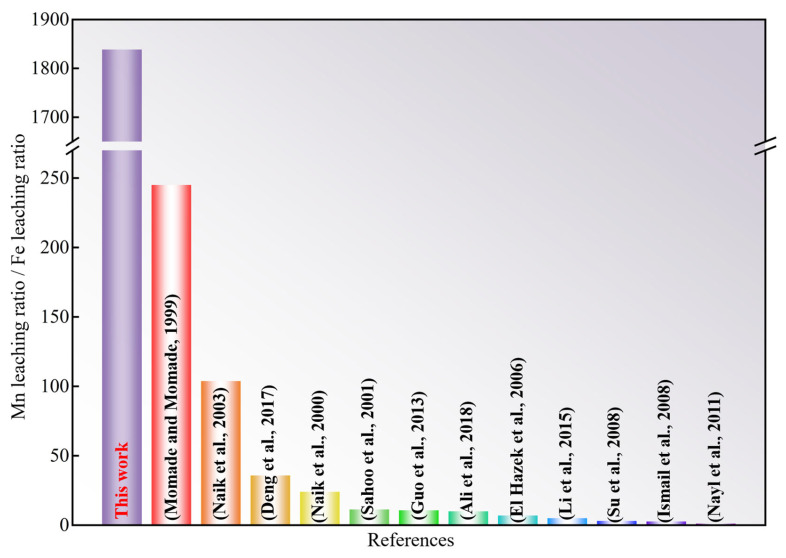
Separation efficiency of manganese and iron in different studies of Fe-Mn ore [10,16,18,20,21,22,29,30,31,32,33,34].

**Figure 12 materials-18-00844-f012:**
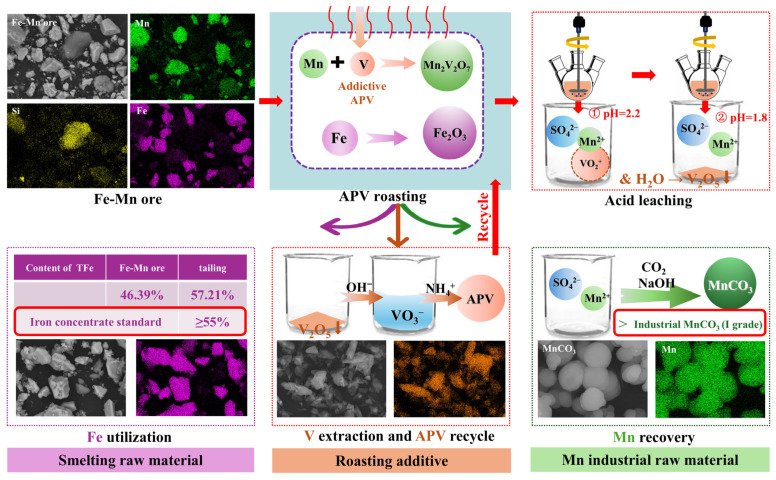
Process flow sheet of Fe-Mn ore utilization based on APV roasting.

**Table 1 materials-18-00844-t001:** Experimental parameters for mixed roasting of Fe-Mn ore and APV.

Variables During Roasting	*n*(MnO_2_)/*n*(V_2_O_5_)	Roasting Temperature (°C)	Holding Time (min)	Heating Rate (°C/min)
Experimental parameters	0; 0.5; 1; 1.5; 1.75; 2; 2.15; 2.25; 2.35; 2.5; 3	300; 400; 500; 600; 700; 800; 850; 900; 950	180	5

**Table 2 materials-18-00844-t002:** Chemical compositions of Fe-Mn ore (wt.%).

Compositions	MnO_2_	Fe_2_O_3_	SiO_2_	Al_2_O_3_	Total
Fe-Mn ore	26.97	66.27	4.63	1.69	99.56

**Table 3 materials-18-00844-t003:** Chemical compositions of vanadium- and iron-containing tailing (wt.%).

Compositions	TMn	TFe	Fe_2_O_3_	MnO_2_	SiO_2_	Al_2_O_3_	V_2_O_5_	Total
V and Fe tailing	2.67	44.01	62.88	4.22	4.73	1.57	19.01	92.41

**Table 4 materials-18-00844-t004:** Chemical compositions of iron-containing tailing (wt.%).

Compositions	TMn	TFe	Fe_2_O_3_	MnO_2_	SiO_2_	Al_2_O_3_	V_2_O_5_	Na_2_O	Total
Fe tailing	4.39	57.21	81.73	6.94	5.15	1.82	0.50	0.067	96.21
Iron concentrate standard	-	≥55 (H_55_)	-	-	≤12 (I class)	-	-	≤0.25	

## Data Availability

The original contributions presented in this study are included in the article. Further inquiries can be directed to the corresponding author.

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
