# Peer review of "New Insight into a Green Process for Iron Manganese Ore Utilization: Efficient Separation of Manganese and Iron Based on Phase Reconstruction by Vanadium Recycle"

_materials, 2025, doi:10.3390/ma18040844_

Round 1
Reviewer 1 Report
Comments and Suggestions for Authors
Dear authors,
This manuscript provides insight into worth studying whether vanadium can serve as a medium for separating manganese and iron in the extraction process of Fe-Mn ore. The subject covered in this article will interest Journal Materials readers but may be accepted for publication only after minor revision. Here are my comments:
Comment 1: English correction is required.
Comment 2: Lines 40-41 –…,, China has abundant manganese reserves, but in which the grade of manganese is low…” - Please rephrase the sentence. The meaning is not satisfactory, because if China has abundant manganese reserves, then the manganese level should not be low. Perhaps it is more appropriate to say that China is plentiful in the number of deposits in which manganese is present but in a small proportion compared to the other elements.
Comment 3: Line 78 - At the end of the introductory section, please explain better the goal and purpose of your research presented in the article and what is it that appears for the first time only in your research. Also, please explain the proposed efficient and green strategy for using Fe-Mn ore in more detail.
Comment 4: Line 79 - Material and methods - In this segment of your article, you should change the order. Namely, within the materials segment, you should state where the ore you used came from. State the locality or company that provided you with those samples. Then, state the manufacturers' names of all the chemicals you used and their purity. After that, you should separate the methods segment, which represents your characterization segment. Only when you have clearly presented the basic data should you continue with the text related to Figure and Table 1 and all other segments within this section.
Comment 5: Lines 165-166 - X-ray Diffraction Analysis (XRD, X’PERT PRO MPD/PW3040, PANalytical B.V. Corporation, Netherlands, Cu Kα radiation from 10° to 80°)…” - Please specify the recording conditions and which database you used for phase identification.
Comment 6: Self-citation is noted, but I believe it is consistent with the text as a whole. Despite this, please check again whether all these references are necessary or whether citing others that address this topic would be more appropriate.

Author Response
Due to the large amount of content in the response draft, we have uploaded it as an attachment.

Reviewer 2 Report
Comments and Suggestions for Authors
Materials
New insight into a green process for iron manganese ore utilization: efficient separation of manganese and iron based on phase reconstruction by vanadium recycle
Review
1.
and the mass ratio of TMn and TFe is 0.37, far 82
What does "T" Mn mean
2.
During leaching, the roasted materials were put in the deionized water with the ratio of 121
liquid to solid (L(deionized water volume)/S(roasted materials mass); mL/g) of 5 until the deionized water 122
most often it is written short, i.e.: L/S=5; ml/g
3.
The tailing was put in the deionized water with the ratio of liquid to solid 146
(L(deionized water volume)/S(tailing mass); mL/g) of
(L/S=5; mL/g)
4.
3.1.1. Effect of n(MnO2)/n(V2O5) 176
n- ??? Please explain what is n, - number of moles? mass?
5.
Fe-Mn ore without addictive could 184
Fe-Mn ore without addictive APV could
6.
Figure 3. XRD and SEM of the roasted materials under different amount of APV. 220
The drawing is very interesting, but it would be possible to separate the diffractograms and map analyses separately, then the SEM-EDS point analysis will be more visible, and the article will benefit from such a solution.
The drawings are cut out imprecisely, there are traces of peaks that need to be removed.
7.
Figure 4.
The same as in 6.
8.
perature according to Eq. (6). MnO2 and Mn7SiO12 are still the main manganese-containing 230
phases. At 500°C, the diffraction peaks of MnO2 are replaced by those of Mn2O3, indicating 231
that MnO2 gradually decomposes to form Mn2O3 as shown in Eq. (7). In addition, diffrac- 232
I suggest that authors insert the equation right after the entry in the text, and not force the reader to search through the text.
with increasing roasting tem- 229perature according to Eq. (6):
(NH4)2V6O16 = 3V2O5 + 2NH3↑+ H2O (6)
9.
3.1.1. Effect of n(MnO2)/n(V2O5) 176
3.2.1. Effect of n(MnO2)/n(V2O5) 263 ????
10.
3.1.2. Effect of roasting temperature 221
3.2.2. Effect of roasting temperature 295
Please sort this out somehow
11.
Figure 6 Effect of roasting temperature on the leaching behaviors of (a) manganese, (b) iron and vanadium 316
Figure 6. Effect of roasting temperature on the leaching behaviors of (a) manganese, (b) iron and vanadium 316
12.
Results show that the TFe content of the iron-bearing tailings is 375
57.21%, which is nearly 11 percentage points higher than the TFe content of Fe-Mn ore. 376
Results show that the TFe content of the iron-bearing tailings is 57.21%, which is nearly 11 percentage points higher than the TFe content of Fe-Mn ore (Table 1).
13.
with some deionized 419 water added.
with some deionized water added to complement ?
14.
g. Hence, more than 99% of vana- 421
dium in the additive APV can be recovered and recycled back into the process of Fe-Mn 422
ore utilization, which can be confirmed that the content of vanadium in the iron-contain- 423
ing tailing is only 0.50%. I
in the entire work review and fill in what % these are: wt.%, at.%, vol.% etc.
15.
Figure 12. Process flow sheet of Fe-Mn ore utilization based on APV roasting. 429
This looks more like a graphic abstract; it would be more advantageous to make a chart, e.g. a Sankey char/flow, showing the distributions of Mn, V, Fe in the individual stages of this extremely interesting process.
Author Response

(The authors gave the same response as above.)

Reviewer 3 Report
Comments and Suggestions for Authors
The manuscript entitled “New insight into a green process for iron manganese ore utilization: efficient separation of manganese and iron based on phase reconstruction by vanadium recycle” presents an efficient route for the separation of Mn and Fe through the application of ammonium polyvanadate (APV). The manuscript is concise, presents extremely interesting results, is well-written, well-structured, and fits within the scope of Materials. However, there are some points that need to be improved for it to be accepted for publication, as follows:
- Table 1 – adjust “compositions.”
- Fig. 1 – include the identification above the peaks and remove the baseline below the diffractogram.
- The XRD phases in Fig. 1 need to be better explained in the text, not just presented. All these mineralogical phases should be described, e.g.: Quartz (SiO2 – JCPDS card no. 00-046-1045).
- In item 2.2.1 – Mixed roasting, the range 0-3 should be included in a table with the studied contents.
- How did the firing process work? It’s not clear: the heating rate was from room temperature to 300-950 °C. Were the samples fired at different temperatures? It is important to describe which temperatures.
- Line 157 – Does 2.2.3 refer to section 2.2.3? If so, include “section 2.2.3” in the text.
- Line 166 – XRD ratio 10-80 °. However, Figures 3 and 4 range between 10 and 70 °. Adjust accordingly.
- Figures 3 and 4 need improvement. There are some markings below n(MnO2)/n(V2O5)=2 and 0.5 that should be removed. Another important point: when including a table in the methodology section with the mixtures, replace “n(MnO2)/n(V2O5)” with M (mixture), such as M0 (No APV), M1, M1.5, M2, M2.25, M2.5, and M3. Include FMV, M2, §, ©, Ñ, •, ¨, and a below the image, along with the Figure titles. This will make the diffractograms cleaner and easier to interpret. Include the JCPDS cards next to each phase in the text when mentioned.
- In Figures 7a, 9a, and 10a, the identification should be placed above the peaks.
Author Response

(The authors gave the same response as above.)

Reviewer 4 Report
Comments and Suggestions for Authors
1- I consider that all the characterization shown in the materials and methods section (Table 1 and Figure 1) should be moved to the results discussion section.
2- EDS composition analysis is not quantitative since it is a semi-quantitative technique, so it is not decisive in chemical composition analysis. This must be complemented with a technique that does provide accurate information.
3- I consider the manuscript to be well structured, except for minor suggestions. What is discussed is well understood, it is relevant and it is a topic that addresses environmental issues.

Author Response

(The authors gave the same response as above.)

Round 2
Reviewer 3 Report
Comments and Suggestions for Authors
After analysis of the authors' corrections, the paper becomes suitable for publication.